# The Benefit of Bimodal Training in Voice Learning

**DOI:** 10.3390/brainsci13091260

**Published:** 2023-08-30

**Authors:** Serena Zadoorian, Lawrence D. Rosenblum

**Affiliations:** Department of Psychology, University of California, Riverside, CA 92521, USA; serena.zadoorian@email.ucr.edu

**Keywords:** audiovisual integration, multisensory processing, talker learning, face perception

## Abstract

It is known that talkers can be recognized by listening to their specific vocal qualities—breathiness and fundamental frequencies. However, talker identification can also occur by focusing on the talkers’ unique articulatory style, which is known to be available auditorily and visually and can be shared across modalities. Evidence shows that voices heard while seeing talkers’ faces are later recognized better on their own compared to the voices heard alone. The present study investigated whether the facilitation of voice learning through facial cues relies on talker-specific articulatory or nonarticulatory facial information. Participants were initially trained to learn the voices of ten talkers presented either on their own or together with (a) an articulating face, (b) a static face, or (c) an isolated articulating mouth. Participants were then tested on recognizing the voices on their own regardless of their training modality. Consistent with previous research, voices learned with articulating faces were recognized better on their own compared to voices learned alone. However, isolated articulating mouths did not provide an advantage in learning the voices. The results demonstrated that learning voices while seeing faces resulted in better voice learning compared to the voices learned alone.

## 1. Introduction

When perceiving speech, listeners not only extract the linguistic message but also information about the talker’s identity. Typically, listeners can learn to recognize the voice of unfamiliar talkers very soon after being introduced to that talker [1]. However, individuals with hearing loss and/or cochlear implant(s) often have difficulty recognizing voices [2,3,4]. Although feedback training has been shown to help train those with good hearing in learning to recognize voices [1,5], this training technique has not been as effective for those with hearing impairments or implants [6]. For example, Cleary and Pisoni [6] found that cochlear implant users had greater difficulty discriminating talkers during test trials compared to normal-hearing individuals after completing a short set of practice trials. The question then arises of whether effective training methods can be designed to help with voice recognition. The goal of the following project was to examine the utility of a multisensory method for learning talker’s voices.

Voice recognition is usually thought to be accomplished by listening for a talker’s specific vocal qualities, such as the fundamental frequency of phonation and breathiness [7,8,9]. However, other recent research has shown that talkers can also be identified through their articulatory style or idiolect—the idiosyncratic way that a talker articulates segments and syllables [9,10,11]. As shown by Remez and colleagues [11], talkers can still be identified even when speech signals do not contain fundamental frequency or timbral properties (e.g., breathiness). Talkers can be recognized from sinewave speech composed of nothing more than three simple sinewaves that track center formant frequencies of a speech utterance. Remez and his colleagues argue that while sinewave speech does not contain the acoustic properties typically thought to support talker identification, the signals do contain talker-specific articulatory information that can also identify a talker. Interestingly, this talker-specific articulatory information is also available in the visual speech modality (for lipreading). In fact, talkers can be recognized from isolated visual speech alone [12]. As shown by Rosenblum and colleagues [12], familiar talkers were recognized at better than chance levels from isolated visible articulatory information (using a point-light method). Research suggests that there is also common talker-specific articulatory information available *across* auditory and visual modalities [11,13]. For instance, Lachs and Pisoni [13] found evidence for cross-modal talker matching using sinewave and point-light talker stimuli to isolate the articulatory information in both modalities.

Finally, a recent study [14] examined whether learning talkers from their articulatory style in one modality could facilitate learning the same talkers in the other modality. Simmons and colleagues [14] observed that learning to recognize talkers from the isolated visible articulatory information of point-light speech facilitated learning of those same talkers from the auditory articulatory information of sinewave speech. The authors interpreted their results as showing that perceivers can learn to use ‘amodal’ talker-specific articulatory information that is available across—and can be shared between—the modalities for talker learning.

Based on these findings, the question arises whether adding visual speech to the auditory signal can facilitate the learning of new voices from auditory-alone information. It is well known that adding visual information facilitates the intelligibility of speech [15,16,17]. There are a small number of studies showing that voice learning can also be facilitated in this manner, e.g., [5,18]; see also [19]. von Kriegstein and colleagues [18] examined this question by performing behavioral and neuroimaging experiments with prosopagnosics and matched controls. Participants either both heard and saw three talkers (voice-face condition) uttering sentences or just heard sentences while being presented with static symbols of different occupations (voice-occupation condition). Participants were later tested on their ability to match the name to the voices, which included both speech and nonspeech (vehicle) sounds. Results indicated that participants were more accurate in recognizing (auditory-only) voices for the talkers presented during the voice-face learning condition compared to those presented in the voice-occupation learning condition. However, this face benefit was not observed in the prosopagnosic patients. Interestingly, a positive correlation between fusiform face area (FFA) activity and face benefit in speaker recognition in the control group was found. This could mean that FFA activity during audio-only talker recognition (for talkers who were part of the voice-face learning condition) might have optimized voice recognition. It is important to note that the study of von Kriegstein et al. [18] used (a) nonface test stimuli (vehicle sounds) that are different from the more common audio-alone control stimuli more typically used in the literature, and(b) the training phase only included a small set of talkers (three).

In a more recent study, von Kriegstein and her colleagues [20] showed that prior audiovisual talker learning facilitates audio-alone voice recognition even when the auditory information is weaker (i.e., speech-in-noise). Their training method was similar to the one discussed above [18]. The authors interpreted their findings as showing that learning to recognize a talker after seeing their face may recruit visual mechanisms, allowing a better recognition of audio-alone voices under challenging listening conditions.

In another study, Sheffert and Olson [5] also attempted to determine whether multisensory speech training would facilitate voice learning. Because our own study is based on their design, details will now be provided on Sheffert and Olson’s [5] method. The study consisted of three phases: familiarization, talker training, and test. During the familiarization phase, participants were presented with 10 talkers uttering words and were provided with their identity (e.g., “That was Tom”). Next, the talker training phase consisted of one of two conditions—audiovisual and auditory-only—in which participants were presented with a set of words spoken by the talkers and were asked to verbally name the talker who presented each word. After each trial, participants received feedback about whether they were correct and, if not, the name of the correct talker. The final phase tested if participants learned the talkers’ voices by presentations of just the voices alone (without accompanying faces). Participants were required to achieve an average of 75% correct voice recognition during this test phase, which was auditory-only, regardless of the training modality (audiovisual or audio-only). If participants failed to achieve an average of 75% correct voice recognition, they were asked to come back the next day to complete another training and test phase. For this reason, the experiment lasted over multiple days.

There are several noteworthy findings from the Sheffert and Olson [5] study. First, results from the training sessions revealed an advantage for learning voices presented audiovisually vs. just auditorily. This is not surprising because, during training, the task in the audiovisual condition could be accomplished by face as well as voice recognition. More impressively, participants who were trained to see the talkers’ faces were significantly more accurate (especially during the first couple of days of training) at identifying the talkers’ voices during the audio-alone test phase compared to those trained with just auditory stimuli. These results are consistent with the findings of von Kriegstein and colleagues [18,20] and suggest that seeing a talker speaking can facilitate the learning of the talker information present in the auditory signal.

While there is evidence that the presence of a face can help perceivers better learn a voice, it is unclear which specific dimensions of facial information allow for this facilitation. Based on the earlier discussion, it could be that the amodal talker-specific information that supports face-voice matching also facilitates voice learning. However, the nonarticulatory information in a face might also help voice learning.

In fact, there is evidence of a perceivable connection between a person’s (static) facial appearance and voice. Mavica and Barenholtz [21] found that the voices of unfamiliar talkers could be matched to *static* images of their faces at better-than-chance levels. In addition to the matching task, participants were asked to rate the faces and voices on physical (e.g., weight, height, age, and attractiveness) and personality (e.g., extraversion, openness, and agreeableness). However, the authors found no obvious relationship between the matching task and these ratings. The authors concluded that participants might have used a set of properties—available in both faces and voices—containing amodal information that helped them group the facial and vocal characteristics.

Related research has shown that correlations exist between individuals’ facial and vocal attractiveness [22] when participants were asked to make independent judgments of static images and voices [23]. These findings show that it is possible that cross-modal attractive information is used to match static images of faces to their voices. Other characteristics such as height, age [24,25,26], and personality traits (e.g., extroversion) [27,28,29] can be inferred from static images and/or recordings of voices as well.

Research has been conducted to examine the relevance of dynamic facial information in matching faces to voices. In a study conducted by Huestegge [30], participants were asked to match a voice to either two static or two dynamic faces. Results showed that the performance between the static and dynamic face-voice matching was similar. This may suggest that the information used by participants to match faces to voices is available in the static faces, thereby indicating that the dynamic information may not offer further advantages.

While dynamic facial information may not always provide additional benefits over static face information in the context of a matching task, it has been shown to play a role in the recognition of familiar voices. In a study carried out by Schweinberger and his colleagues [31,32], participants were simultaneously presented with a face (dynamic or static) along with either corresponding or noncorresponding voices (both familiar and unfamiliar). Participants were asked to judge the familiarity of the *voice*. Results showed that recognizing the voices of familiar talkers was better when the voices were paired with their correct corresponding faces. This improvement was stronger with the (corresponding) dynamic than static faces (with either dynamic or static faces; also see [33,34]). Importantly, this improvement was larger for faces presented in a time-synchronized- vs. unsynchronized manner, suggesting that synchronization of dynamic information may play a role in identifying familiar speakers.

However, a recent study has shown that unfamiliar voices (compared to novel voices) were recognized better (i.e., categorized as new) on their own after being paired with static faces [35]. In that study, EEG recordings during both the learning and voice recognition phases also revealed an early interaction between voice and face processing areas. Thus, there is evidence that static faces can be matched to voices and that, in some contexts, static faces can facilitate the identification of voices as unfamiliar. However, it is unclear if static face information alone might help facilitate the *learning* of novel voices without the presence of visible articulation. This question will be tested in the following study.

It should also be mentioned that while a number of studies have shown face facilitation of voice recognition, some studies have reported that the presence of faces can *inhibit* voice recognition [1,36]. In what is known as the ‘facial overshadowing effect’, e.g., [1,36,37], the presentation of the face has been thought to direct attention away from the voice, thereby impairing performance during voice-only recognition [1,36,37,38]. In fact, this inhibitory effect has been shown to be stronger with the presence of direct eye gaze in the face, which may be linked to capturing more attention [39]. Although it is true that in the present study, both the articulating and static face training conditions included a direct gaze of the speakers, it is important to note that the training included other characteristics that may have outweighed the effect. For example, in the present study, participants were presented with audio-alone probe trials during training to ensure they were learning the voices. Additionally, during the test phase, participants were asked to identify the talkers (compared to categorizing voices as old/new).

However, it is important to note that the procedures involved in studies showing the ‘facial overshadowing effect’ focused on voice *recall* and not voice identification. Participants in these studies, i.e., [1,36,37,38], were asked to study a set of talkers’ voices and were then tested on whether they could recall hearing a particular voice. The findings of these studies run contrary to those that focused on voice *identification*—learning to identify particular talkers based on vocal characteristics—discussed earlier [5,18], which showed an advantage of the presence of the talkers’ faces during training. The current study will examine this latter scenario.

### Purpose of the Current Study

The purpose of the current study was to examine the effect of multisensory training in facilitating voice identity learning. We were interested in examining the role of visible articulatory information and static face information in learning to recognize audio-only voices. Our method involved training participants using audio-only or audiovisual stimuli and had all participants tested on (audio-only) voice learning. Thus, in the current experiment, we attempted to confirm the findings of Sheffert and Olson [5] using a modified procedure and test the benefit of visible articulatory information more directly. For this reason, we added two new conditions to the Sheffert and Olson [5] paradigm for training voice recognition in which we (a) presented static images of the speakers along with their audio files and (b) isolated the mouth area of the speakers as they were articulating sentences. It should be mentioned that Sheffert and Olson [5] conducted an informal follow-up control experiment to evaluate the importance of mouth movements during their audiovisual training sessions. Using a small group of participants, they presented their training audiovisual stimuli with the talkers’ mouths occluded (including the lower cheeks, the jaw, and the nose) on the presentation screen. Results showed that these reduced face stimuli failed to facilitate voice learning over audio-only training conditions. Based on this result, Sheffert and Olson [5] concluded that the advantage of audiovisual training observed in their main experiment was likely due to the presence of visible articulatory information rather than nonarticulatory facial identity information.

While the data from this informal follow-up experiment are useful, the conclusions must be considered preliminary. It could be, for example, that occluding the mouth in this manner removes not only articulatory information but also nonarticulatory (e.g., mouth and chin features) information that may, in principle, be useful for voice learning. For these reasons, we decided to more directly test the possible facilitative utility of both static facial and articulatory information for voice learning by including both *static face* and *isolated (moving) mouth* visual stimuli in our tests. Four hypotheses were made based on the assumption that the multisensory voice learning facilitation observed in the prior research was based on the availability of visible articulatory information. First, if participants can use visible talker-specific articulatory information to facilitate voice learning, then participants in the (full-face) articulating face condition would significantly outperform those in the audio-only condition when identifying the voices during the (audio-only) test phase. Second, if the visible articulatory information is *sufficient* to facilitate voice learning, then those trained with isolated mouth stimuli should also outperform those with audio-only training in the test phase. Third, performance after training with both articulating face and mouth-only stimuli should be better than that after training with static face stimuli. Finally, if visible articulatory information provides the *necessary* information for multisensory facilitation of voice learning, then performance in the static face condition should be no better than that in the audio-only condition.

## 2. Method

Due to the COVID-related protocol, the experiment was administered online through Pavlovia.org [40]. Due to the difficult and unique nature of the stimuli and task, the decision was made to have an experimenter monitor each participant over a Zoom connection, which limited the number of participants we could recruit online. All participants were asked to share their screen over Zoom with the experimenter and were required to use earphones. The experiment took about an hour and a half to complete, and participants received course credit for their participation.

### 2.1. Participants

A total of ninety-six participants (*Mean age* = 19.71, *S* = 2.26) were included in the study. The number of participants tested was based on (a) a similar number used in related experiments e.g., [5], and (b) the limitations of running monitored participants online during COVID quarantine. As per a reviewer’s suggestion, a post hoc power analysis was conducted using the R package “simr” [41]. The effect size was calculated using the doTest function from the “lmertest” package [42]. The effect size calculation was based on 50 simulations using the *z*-test, and the alpha level was set to 0.05. The effect size for the articulating face condition was equal to 0.61 with an observed power of 70% (CI: 55.39, 82.14). The effect size for the static face condition was equal to 0.43 with an observed power of 38% (CI: 24.65, 52.83). Lastly, the effect size for the isolated articulating mouth condition was equal to −0.019 with an observed power of 8% (CI: 2.22, 19.23). Twenty-four (14 female) participants were randomly assigned to the *articulating face* condition, twenty-four (17 female) individuals were randomly assigned to the *isolated mouth* condition, twenty-four (18 female) individuals were randomly assigned to the *static face* condition, and another twenty-four (19 female) participants were randomly assigned to the *audio-only* condition. Eighty-three of these individuals were native English speakers. Out of the thirteen non-native-speaking participants (four in the *articulating face*, three in the *isolated mouth*, two in the *static face*, and four in the *audio-only* conditions), ten were native Spanish speakers. The rest did not report their native language on the questionnaire (the critical analysis (comparing the test means between the four conditions) was repeated after excluding the non-native speakers, and the analysis yielded the same results). All participants reported having normal hearing and normal to corrected vision.

### 2.2. Materials

Stimuli were recordings of 4 female and 6 male native English talkers. Talkers were video-recorded speaking four utterances of ten different sentences (forty recordings each) taken from the Bamford–Kowal–Bench sentence list [43]. Talkers were instructed to speak as naturally as possible and were recorded using a SONY DRC-TRV11 (Tokyo, Japan) camcorder. The recordings were then digitized using a digital video camera (DRC-TRV11) and Final Cut Pro X (2011). To create the isolated mouth stimuli, the fully illuminated faces were edited in Final Cut Pro X (2011) by adding a digital black mask to all areas of the face except for an oval around the mouth region of each talker (see Figure 1). The final stimuli for the isolated mouth condition consisted of a small oval surrounding the talkers’ mouths against a black background. These stimuli only retained the articulatory information on the mouth that is typically considered the most important for lipreading. To create the static face stimuli, the fully illuminated faces were edited in iMovie (version 10.2.3) by selecting a single frame for each stimulus file (see Figure 1). The single frames were selected based on the talkers showing a neutral mouth position. The final static image stimuli consisted of a static photograph of the talker that was presented for the length of their corresponding audio file. Lastly, the stimuli for the audio-only condition were created by resaving the audiovisual files as audio-alone files.

### 2.3. Procedure

The experiment consisted of three phases: familiarization, training, and test. The familiarization and test phases were the same for all participants, so only the training phase varied across the four conditions. It is important to note that all participants in all conditions were instructed to concentrate on the learning talkers’ *voices* (despite seeing their faces in three of the conditions).

***Familiarization Phase***: During the familiarization phase, participants were presented with the talker’s name (e.g., “This is Liz”) followed by two repetitions of an auditory-alone utterance of that talker. After hearing the two presentations, the talker’s name was again presented (e.g., “That was Liz”). There were two presentations of each of the talkers (using the same utterance) for a total of 20 randomized trials for the familiarization phase. The utterance chosen was different for each talker. Participants did not make any responses during this phase.

***Training Phase:*** During the training phase, the main training trials involved presentations of two utterances (of the four recorded utterances) of all ten sentences of each talker. This created a total of 200 main training trials whose format depended on the assigned presentation condition. In addition, the training phase also included 40 ‘probe’ trials that were presented in audio-alone format. These probe trials occurred after every five training trials and were used to carry out the following: (a) evaluate whether participants were learning the *voice* of the talkers; (b) remind participants that their goal was to learn the talkers’ voices even if they were able to see their faces on the training trials. The addition of the 40 probe trials created a training phase consisting of 240 total trials. On a given training trial (both main training and probe), a single utterance was presented, and participants were asked to press 1 of 10 keys on their keyboard to indicate which talker they believed they had heard. Each talker was assigned a number that corresponded with the numbers on the keyboard. Participants were able to see the names and corresponding numbers on the bottom of their screen as soon as each utterance presentation ended and until they made a response. After making a response, feedback was given such that participants were told if they were correct and who the speaker was. During the main training trials, participants assigned to the *articulating face* condition saw the talkers’ articulating faces in synchrony with each audio sentence. Those assigned to the *isolated mouth* condition were presented with the isolated mouths of the talkers presented synchronously with each heard utterance. Participants in the *static face* condition were presented with single still images of the talkers presented for the duration of each heard sentence. Lastly, participants assigned to the *audio-only* condition were presented only with the talkers’ audio voices uttering the sentences. Recall that all probe trials for all condition groups were always audio-alone presentations.

***Test Phase***: During the test phase, all participants were presented with two new audio-alone utterances of the ten sentences of each talker. This phase consisted of 200 randomized trials, which were presented in a different order than the training phase. On a given trial, participants were presented with a single utterance auditorily, and they were asked to identify the talker they believed they heard by pressing a key on their keyboard. The talkers’ assigned number keys were the same as in the training phase. Similar to the training phase, participants saw the names and corresponding numbers on the bottom of their screen after each utterance ended and until a response was made. Unlike the training phase, feedback was not provided during the test phase. Participants completed the experiment on a laptop or desktop computer using Chrome, Safari, Microsoft Edge, or Firefox and were asked to wear headphones throughout the experiment. The average screen size used by participants was 13.3″ (89 participants used smaller screens ranging from 11.6″ to 16″ and 7 used bigger screens ranging from 21.5″ to 34″). About 65% of our participants listened to the stimuli using earphones (e.g., Airpods and Sony), while the remainder listened using over-the-head headphones. Each participant adjusted the sound to a comfortable listening level.

## 3. Results

### 3.1. Test Phase

To examine the effect of the various training methods in facilitating voice learning (audio-only), we first performed a one-sample *t*-test to test against chance performance (10%) and then analyzed the data using mixed-effect logistic regressions and analysis of variance (ANOVA; significance was set at *p* < 0.05). A one-sample *t*-test revealed a significant result indicating that participants in the *articulating face* condition identified talkers at a better than chance (10%) level (*M* = 0.65, *S* = 0.19), *t*(23) = 14.20, *p* < 0.001, *Cohen’s d* = 2.90. Talker identification rate varied significantly across talkers, *F*(9) = 24.72, *p* < 0.001, *np*^2^ = 0.04 (see Figure 2). All talkers were identified at above chance levels at a Bonferroni-corrected *α* = 0.005. Another one-sample *t*-test revealed a significant result indicating that participants in the *isolated mouth* condition identified talkers at a better than chance (10%) level (*M* = 0.54, *S* = 0.15), *t*(23) = 14.61, *p* < 0.001, *Cohen’s d* = 2.99. Talker identification rate again varied significantly across talkers, *F*(9) = 34.23, *p* < 0.001, *np*^2^ = 0.06 (see Figure 2). Yet, talkers were identified at above chance levels at a Bonferroni-corrected *α* = 0.005.

Next, another one-sample *t*-test showed that those in the *static face* condition identified talkers at better than chance levels (*M* = 0.63, *S* = 0.19), *t*(23) = 13.34, *p* < 0.001, *Cohen’s d* = 2.72. The Bonferroni-corrected (*α* = 0.005) analyses revealed that talker identification varied significantly across talkers, *F*(9) = 6.31, *p* < 0.001, *np*^2^ = 0.01. A final set of one-sample *t*-tests revealed that participants in the *audio-only* condition identified talkers at a better than chance level (*M* = 0.55, *S* = 0.13), *t*(23) = 16.74, *p* < 0.001, *Cohen’s d* = 3.41. Talker identification rate varied significantly across talkers, *F*(9) = 43.65, *p* < 0.001, *np*^2^ = 0.08 (see Figure 2). The post hoc *t*-test analyses revealed that all talkers were identified at a better than chance level at a Bonferroni-corrected *α* = 0.005.

A set of logistic mixed-effect models was used to compare the test means between the four conditions (see Figure 3). All of the reported analyses treated the condition as a fixed effect and participants and talkers as random effects. The first analysis revealed a significant difference between the articulating face and audio-only conditions (*β* = 0.57, *SE* = 0.24, *z* = 2.36, *p* = 0.018, CI: 0.09–1.04) favoring the *articulating face condition* (65% vs. 55%, respectively). Next, no significant difference was found between the isolated mouth and audio-only conditions (*β* = −0.02, *SE* = 0.24, *z* = −0.08, *p* = 0.936, CI: −0.50–0.46) (54% vs. 55%, respectively). There was no significant difference between the test means of the static face and audio-only conditions (*β* = 0.43, *SE* = 0.24, *z* = 1.78, *p* = 0.076, CI: −0.05–0.90).

A significant difference was found between the test means of the isolated mouth and articulating face conditions (*β* = −0.59, *SE* = 0.24, *z* = −2.44, *p* = 0.015, CI: −1.06–−0.11) favoring the *articulating face* condition (53% vs. 65%, respectively). However, no significant difference was found between the static face and articulating face conditions (*β* = −0.14, *SE* = 0.24, *z* = −0.59, *p* = 0.558, CI: −0.62- 0.34) (63% vs. 65%, respectively). Moreover, there was no significant difference between the isolated mouth and static face conditions (*β* = 0.45, *SE* = 0.24, *z* = 1.85, *p* = 0.064, CI: −0.03–0.92). Additionally, there was a variability in both the talkers (*var* = 0.31, *S* = 0.56) and participants (*var* = 0.67, *S* = 0.82). A final Analysis of variance revealed a significant interaction between the talkers and conditions *F*(27) = 6.65, *p* < 0.001, *np*^2^ = 0.01.

### 3.2. Training Phase: Probe Trials

In addition to the test trials, we also were able to examine how fast participants in different conditions were learning to recognize the voices in the training phase. As mentioned previously, probe trials were audio-only trials included during the training phase of the four conditions. These trials were originally designed to draw the attention of participants in the audiovisual training conditions (articulating face, isolated mouth, and static face) to learn the voices and not simply the faces. However, these trials could also be used to track how quickly participants learned the voices based on the training condition in which they participated. We analyzed the probe data by performing a set of post hoc mixed-effect logistic regression analyses examining the difference in the overall mean performance. In order to examine the learning rate, we also performed post hoc slope analyses comparing the conditions.

A post hoc mixed-effect logistic regression model compared the overall magnitude of the audio-alone probe trials across conditions (see Figure 4). In the analysis, conditions were added as a fixed effect, and participants and talkers as random effects. Results showed a significant difference between the articulating face and audio-only conditions (*β* = 0.46, *SE* = 0.16, *z* = 2.84, *p* = 0.005, CI: 0.14–0.78) favoring the *articulating face* condition (53% vs. 43%, respectively). Next, a significant difference was not observed between the isolated mouth and audio-only conditions (*β* = −0.17, *SE* = 0.16, *z* = −1.04, *p* = 0.297, CI: −0.49–0.15; 39% vs. 43%, respectively). Similarly, no significant differences were found for the static face and audio-only conditions (*β* = 0.05, *SE* = 0.41, *z* = 0.13, *p* = 0.894, CI: −0.77–0.87; 46% vs. 43%, respectively). The next analysis compared the articulating face and isolated mouth condition probe trials, finding a significant difference (*β* = −0.63, *SE* = 0.16, *z* = −3.88, *p* < 0.001, CI: −0.95–−0.31) favoring the articulating face condition (53% vs. 39%, respectively). However, no significant differences were found between the articulating face and static face conditions (*β* = −0.40, *SE* = 0.41, *z* = −0.98, *p* = 0.325, CI: −1.23–0.41; 53% vs. 46%, respectively). Similarly, no significant differences were found between the isolated mouth and static face conditions (*β* = 0.22, *SE* = 0.41, *z* = 0.54, *p* = 0.586, CI: −0.60–1.04; 39% vs. 46%, respectively). Lastly, the random effects results indicated a variability for both participants (*var* = 0.20, *S* = 0.45) and talkers (*var* = 0.18, *S* = 0.42).

To examine the learning rates, we ran another set of regression models in which condition, probe trials, and the interaction between condition and probe trials were treated as fixed effects, and participants and probe talkers were included as random effects. Accuracy on the probe trials was included as the dependent variable. The analysis revealed a significant main effect of the probe trials (*β* = 0.05, *SE* = 0.01, *z* = 5.49, *p* < 0.001, CI: 0.03–0.06). The analysis also revealed a significant difference in the learning rate of those who were trained with articulating faces compared to those trained auditorily (*β* = 0.02, *SE* = 0.01, *z* = 2.79, *p* < 0.01, CI: 0.01–0.04), favoring the articulating face condition. However, a significant difference was not observed between the isolated mouth and audio-only conditions (*β* = 0.01, *SE* = 0.01, *z* = 1.50, *p* = 0.13, CI: −0.003–0.03). There was no significant difference between the static face and audio-only conditions (*β* = −0.04, *SE* = 0.02, *z* = −1.71, *p* = 0.087, CI: −0.08–0.01).

Additionally, a significant difference was not observed between those who were trained with articulating faces compared to those trained with the isolated mouths (*β* = −0.01, *SE* = 0.01, *z* = −1.27, *p* = 0.203, CI: −0.03–0.01). However, a significant difference between the articulating face and static face conditions was found (*β* = −0.06, *SE* = 0.02, *z* = −2.87, *p* < 0.01), favoring the articulating face condition. Similarly, the analysis revealed a significant difference between the isolated mouth and static face conditions (*β* = −0.05, *SE* = 0.02, *z* = −2.33, *p* = 0.020) favoring the static face condition. Lastly, the random effects results revealed variability within both the participants (*var* = 0.24, *S* = 0.49) and the probe talkers (*var* = 0.14, *S* = 0.38).

### 3.3. Training Phase: Training Trials

Lastly, we examined recognition performance for the nonprobe training trials. Because for three of the conditions, these trials involved audiovisual presentations, and performance on these trials does not represent just voice learning, per se. While the participants were told to concentrate on learning the voices, it is likely that they used the facial information to respond to these trials. Thus, the results of this analysis necessarily inform less about patterns of voice learning. The training analyses examining performance for the *articulating face* condition against chance level only included the trials with articulating faces and not the audio-only probe trials. A one-sample *t*-test revealed that participants in the *articulating face* condition identified talkers at a better than chance (10%) level during the training phase (*M* = 0.87, *S* = 0.10), *t*(23) = 37.93, *p* < 0.001, *Cohen’s d* = 7.74. Talker identification rate varied significantly across talkers, *F*(9) = 5.19, *p* < 0.001, *np*^2^ = 0.01 (see Figure 5). Based on the Bonferroni-corrected (*α* = 0.005) analyses, all talkers were identified at above chance levels.

Similarly, a one-sample *t*-test revealed that participants in the *isolated mouth* condition identified talkers at a better than chance (10%) level during the training trials (*M* = 0.47, *S* = 0.12), *t*(23) = 15.38, *p* < 0.001, *Cohen’s d* = 3.14. Talker identification rate varied significantly across talkers, *F*(9) = 16.700, *p* < 0.001, *np*^2^ = 0.300 (see Figure 5). Based on the Bonferroni-corrected (*α* = 0.005) analyses, all talkers were identified at above chance levels. Next, another one-sample *t*-test showed that those in the *static image* condition identified talkers at better than chance levels (*M* = 0.79, *S* = 0.15), *t*(23) = 20.27, *p* < 0.001, *Cohen’s d* = 4.13. The Bonferroni-corrected (*α* = 0.005) analyses revealed that talker identification varied significantly across talkers, *F*(9) = 6.31, *p* < 0.001, *np*^2^ = 0.01 (see Figure 5). A final one-sample *t*-test revealed that participants in the *audio-only* condition identified talkers at a better than chance level (*M* = 0.46, *S* = 0.10), *t*(23) = 16.80, *p* < 0.001, *Cohen’s d* = 3.42. Talker identification rate varied significantly across talkers as well, *F*(9) = 25.94, *p* < 0.001, *np*^2^ = 0.05 (see Figure 5).

According to the post hoc *t*-test analyses, all talkers were identified at above chance levels at a Bonferroni-corrected *α* = 0.005. Logistic mixed-effect models were used to compare the training means across the four conditions (see Figure 6). We compared the training means (excluding the probe trials) of all conditions against the audio-only condition by treating the condition as a fixed effect and participants and talkers as random effects. This analysis revealed a significant difference between the articulating face and audio-only training methods (*β* = 2.30, *SE* = 0.20, *z* = 11.23, *p* < 0.001, CI: 1.90–2.71) favoring the *articulating face* condition (87% vs. 46%, respectively). However, a significant difference was not observed in the isolated mouth (*β* = 0.08, *SE* = 0.20, *z* = 0.41, *p* = 0.683, CI: −0.32–0.48) condition when compared to the audio-only condition (47% vs. 46%, respectively).

Similar to the articulating face condition, there was a significant difference in the static face (*β* = 1.69, *SE* = 0.20, *z* = 8.36, *p* < 0.001, CI: 1.29–2.10) condition when compared to the audio-only condition favoring the *static face* condition (79% vs. 46%, respectively). Next, there was a significant difference between the articulating face and isolated mouth training methods (*β* = −2.22, *SE* = 0.21, *z* = −10.82, *p* < 0.001, CI: −2.63–−1.82) favoring the *articulating face* condition (87% vs. 47%, respectively). Similarly, there was a significant difference between the articulating face and static face training methods (*β* = −0.61, *SE* = 0.21, *z* = −2.93, *p* = 0.003, CI: −1.02–−0.20) favoring the *articulating face* condition (87% vs. 79%, respectively). Additionally, there was a significant difference between the isolated mouth and static face training method (*β* = 1.61, *SE* = 0.20, z = 7.94, *p* < 0.001, CI: 1.21–2.02) favoring the *static face* condition (47% vs. 79%, respectively). Lastly, variability was found both across the talkers (*var* = 0.11, *S* = 0.33) and participants (*var* = 0.46, *S* = 0.68). A final Analysis of Variance was conducted to examine the interaction between the training conditions and training talkers, which revealed a significant interaction *F*(27) = 8.93, *p* < 0.001, *np*^2^ = 0.01.

## 4. Discussion

The current study examined the use of multisensory training in talker learning. We were interested to see if the presence of visible speech, in addition to audible speech, would facilitate voice learning of unfamiliar talkers. Participants in the current study were trained to recognize talkers either when seeing their full faces, isolated mouths, static images of their faces, or just hearing their voices alone.

### 4.1. Do Faces Facilitate Voice Learning?

Previous research has shown that voices learned with articulating faces are learned better than voices learned without faces [5,18]; see also [19]. However, it is unclear what aspects of facial information facilitate voice learning. To address this question, we first confirmed the findings of Sheffert and Olson [5] by using a modified methodological procedure. As mentioned previously, some of the key methodological differences between the current study and the one conducted by Sheffert and Olson [5] were as follows: (a) using sentences instead of words; (b) including more talkers during our training phase; (c) including the audio-only probe trials to ensure that those who were presented with the faces were learning the voices and not just the faces; lastly, (d) our training session lasted for a shorter duration compared to Sheffert and Olson’s [5] study, which lasted for multiple days.

Despite these methodological differences, the results of our experiment replicated those of earlier studies in showing that participants trained with articulating faces (65%) were significantly more accurate at identifying the audio-only voices compared to those trained just auditorily (55%). Additionally, the post hoc analysis examining the learning rate suggests that those trained with the articulating faces learned the audio-only probe voices (presented during training) at a faster rate compared to those trained auditorily. Thus, our results clearly show face facilitation of voice learning: a finding supportive that the FFA activity previously observed during audio-only talker recognition may optimize voice recognition [18].

Recall that Sheffert and Olson [5] speculated that their results suggested that the talker-specific articulatory style information available in their stimuli facilitated the learning of voices. As additional support, their informal follow-up experiment showed that restricting the articulatory information in their face displays reduced voice learning to the same level as their auditory-alone condition. However, our results may portray a somewhat different story about the importance of visible articulatory information in learning voices. First, while our full articulating face training condition showed consistent facilitation of voice learning, this facilitation was not statistically different from that of the static face condition for the test phase performance. In addition, this measure also showed that while the static face condition was not significantly more effective than the audio-alone training condition, it did show a trend in that direction (63% vs. 55%, *p* = 0.076). Turning to the training phase of the experiment, a significant difference was not observed in the mean performance during the (audio-alone) probe trials in the articulating face and static face conditions. However, only the former was significantly different from the performance during the audio-alone condition training. In addition, the *rate* at which voice learning seemed to occur during the training trials was greater than that of the static condition, based on slope analyses. This learning rate during training for the articulating face condition was also significantly greater than that for the audio-alone condition, while the static face condition was not significantly different than for the audio-alone condition.

Thus, while the results show a clear benefit for articulating faces, there are hints in our data that static faces may also have some benefit, albeit one that is less consistent. In fact, this interpretation is consistent with other findings in the literature, e.g., [31]. The implications of this interpretation are discussed in the following sections.

With regard to the salience of articulatory information for voice learning, the most surprising findings concern the isolated (moving) mouth training condition. While it was hypothesized that this condition would be as effective as the articulating face condition, assuming that articulatory information is most important in voice learning, just the opposite was observed. Based on all measures (e.g., training and test phase data), performance with the isolated moving mouth never facilitated voice learning over the audio-alone condition. In fact, this lack of difference between the mouth and audio-alone conditions was true even for the nonprobe training trials, suggesting that the isolated mouth provided no additional recognition information even when it was present during the training. The implications of this finding will be discussed in the following sections.

### 4.2. Is Visible Articulatory Information Sufficient to Facilitate Voice Learning?

As mentioned previously, those trained with articulating faces learned the voices better compared to those trained auditorily, a finding consistent with other research, e.g., [5,18]; see also [19]. This could suggest the importance of the visible articulatory information available in the articulating faces, which facilitates voice learning. However, the other condition which contained visible articulatory information—the isolated mouth condition—*failed* to facilitate voice learning. This condition was created by isolating the mouths of the talkers and was partly motivated by the informal control experiment conducted by Sheffert and Olson [5]. Recall that in their experiment, participants were presented with just the upper half of the talkers’ faces (no-mouth condition), which excluded the talkers’ mouth, jaw, nose, and lower cheeks. As their results indicated, a significant difference was not observed between these no-mouth and audio-only conditions. Sheffert and Olson [5] interpreted this finding to suggest that seeing the lower part of the face provides visible talker-specific articulatory information that improves voice learning. With this in mind, we reasoned that by conversely presenting the isolated mouth, the facilitation of voice recognition would be comparable to the articulating face condition. However, our results indicated that this was not the case and, in fact, revealed no significant differences between the isolated mouth and audio-only in both the training and test phases of our experiment. It is unclear why the isolated mouth condition failed to facilitate voice learning. The mouth-only stimuli only displayed a very small oval around the mouth area (see Figure 1). It is possible that these stimuli did not capture enough important articulatory information present outside the mouth area. It is known that important visible articulatory information is present throughout the face, e.g., [44,45]. To test the possibility that the mouth-only condition provided too little articulatory information, follow-up studies can be conducted in which additional articulatory information is visible, including movements of the jaw and cheeks.

Another possibility is that the visibly unusual nature of the mouth-only stimuli may have been distracting to the degree that it prevented sufficient attentional resources from being used for good voice encoding. In fact, this possibility is similar to that used to explain the facial overshadowing effects described earlier, e.g., [1,36,37]. Thus, during training, the mouth-only stimuli could have distracted from the sufficient encoding of the voices, thereby canceling out any advantage provided by the available talker-specific articulatory information. This possibility is consistent with the observation that during the training phase of the experiment, no advantage was observed for the nonprobe trials in which the presence of the mouth in the stimuli could have acted as extra information to allow for recognition of the rate of learning (as it did for the other audiovisual conditions). Future research can be designed to test whether the mouth-only condition induced something like a facial overshadowing effect. Additionally, a stimulus condition, which includes the entire lower half of the face (complementary to that used by Sheffert and Olson [5] might work better as a test of articulatory information. A similar approach can be taken using a point-light technique, e.g., [14].

Because our articulating face condition also contained static information, it is impossible for us to conclude that visible articulatory information is sufficient to support audiovisual facilitation of voice learning. While the articulating face condition did support somewhat better facilitation than the static face condition, we cannot rule out that this advantage was due to there being *more* static information in the articulating face displays (for a similar argument, see [12,46]). The static condition contained a single frame of the face, while the articulating face contained up to 90 frames of the face (30 frames/s) in different configurations. Future research using point-light or modified articulating face stimuli can be designed to further examine whether the articulatory information useful for talker identification and cross-modal matching is also useful for voice learning facilitation. As stated, the findings of Simmons et al. [14], that experience learning to recognize talkers from point-light speech facilitated learning of those same talkers from auditory speech, suggests that this possibility is still viable.

### 4.3. Is Static Face Information Sufficient to Facilitate Voice Learning?

Individuals trained with static faces performed as well (statistically) as participants trained with articulating faces during the test phase. These participants also showed a trend toward performing better than those trained with audio-alone stimuli. However, our probe trial results also show that static faces may not be as effective as articulating faces with regard to voice learning rate.

Future research can be focused on whether the presence of static faces could facilitate voice learning in some contexts. As stated, there is evidence for voice-relevant information in static faces. Static images of talkers can be matched to their voices at better than chance levels [21,26]. Participants may be able to make these matches based on visible talker features such as height, weight, or age and their relationship to acoustic features, e.g., [24,25,26,47]. Additionally, information about both personality traits [24,27,29,48] and attractiveness [22,23] can be inferred from both photographs and voice recordings of unfamiliar talkers, potentially supporting successful matching. Regardless, the fact that there is information in static faces suggestive of voice characteristics could help facilitate voice learning. Future research in our laboratory will explore this issue.

## 5. Conclusions

To conclude, voices paired with articulating faces were learned better and faster compared to the voices heard on their own. Additionally, participants trained with static faces performed no worse than those trained with articulating faces but not statistically better than those trained with voices on their own. However, the isolated mouths did not provide any benefits in learning voices.

Ultimately, the findings of the current study have practical implications, especially for those with moderate hearing loss or cochlear implant(s) who have difficulty identifying the voices of talkers [3,4]. Therefore, future research should use the same training methodology to train individuals with hearing loss and cochlear implant patients. The results of this multisensory training might help those individuals to recognize voices better, which may ease their everyday interactions. The findings can also ultimately help researchers design telecommunications systems in order to help those with hearing and language disorders.

## Figures and Tables

**Figure 1 brainsci-13-01260-f001:**
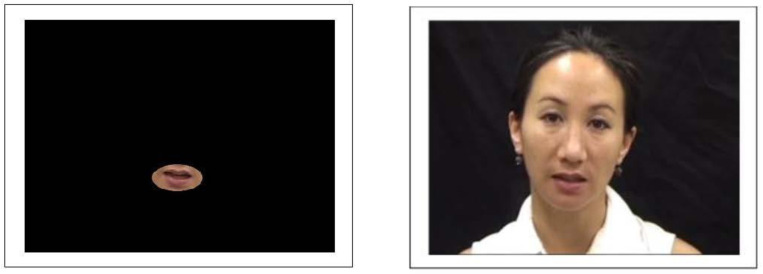
Examples of stimuli: The image on the left is a still image of a sample stimulus used during the isolated mouth-only training condition. The image on the right is a sample stimulus used during the static face training condition.

**Figure 2 brainsci-13-01260-f002:**
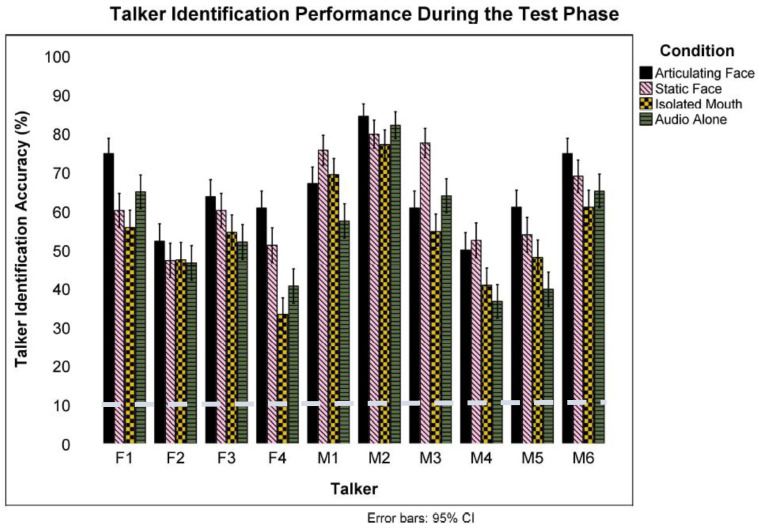
Talker identification performance across all training conditions during the test phase. Error bars indicate the error of the mean. The broken line indicates chance performance (10%).

**Figure 3 brainsci-13-01260-f003:**
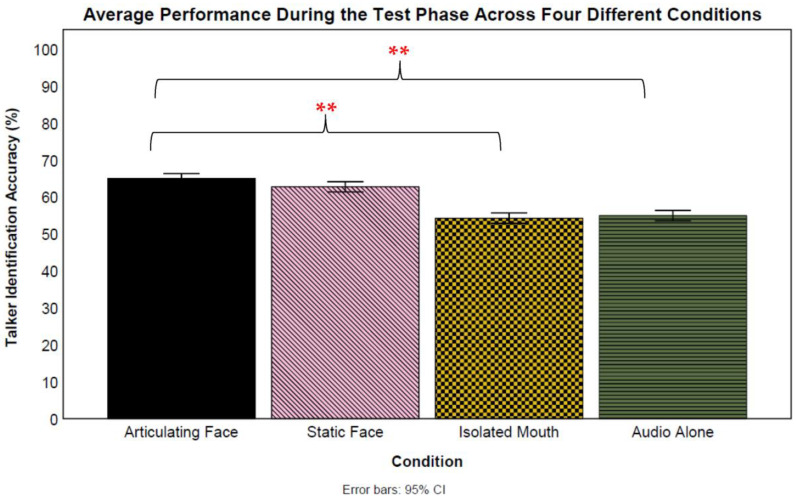
Overall talker identification accuracy during the test phase for all training groups. The error bars indicate standard error of the mean. ** *p* < 0.01.

**Figure 4 brainsci-13-01260-f004:**
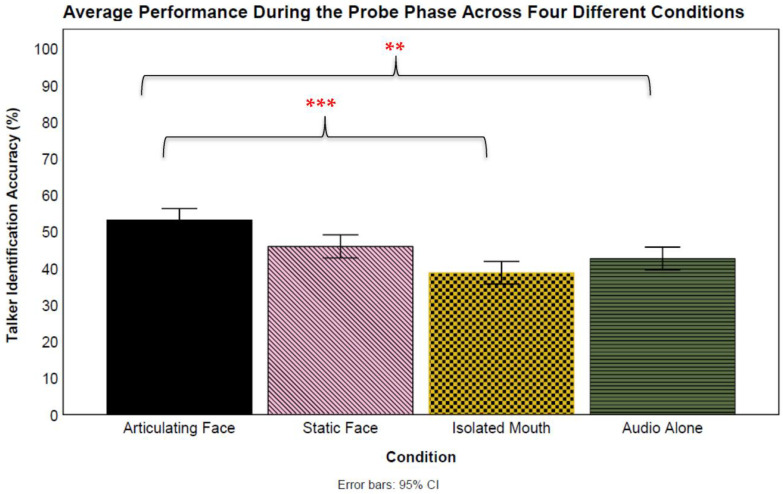
Overall talker identification accuracy during the probe phase for all training groups. The error bars indicate standard error of the mean. ** *p* < 0.01 and *** *p* < 0.001.

**Figure 5 brainsci-13-01260-f005:**
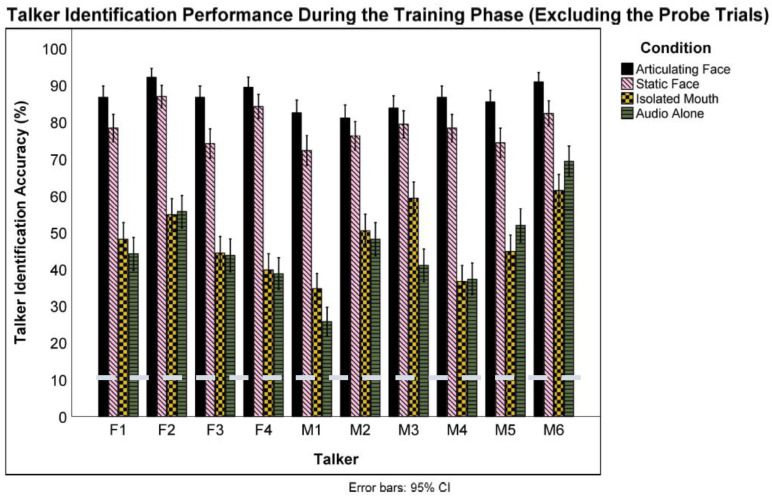
Talker identification performance across all training conditions during the training phase (excluding the probe trials). Error bars indicate the error of the mean. The broken line indicates chance performance (10%).

**Figure 6 brainsci-13-01260-f006:**
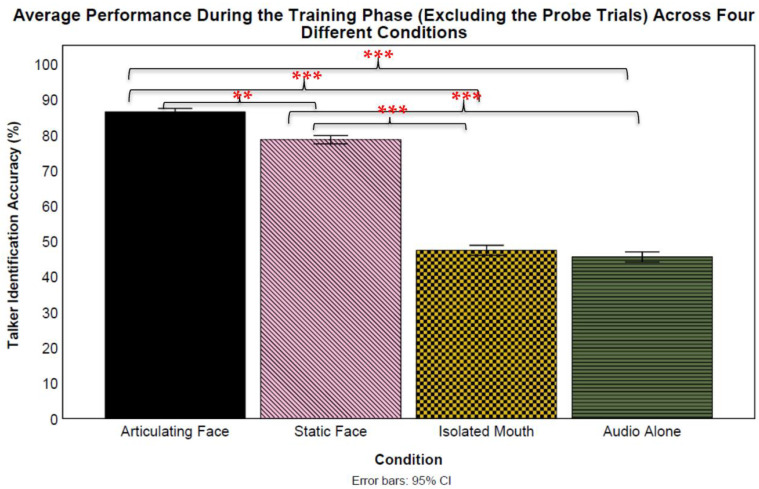
Overall talker identification accuracy during the training phase (excluding the probe trials) for all training groups. The error bars indicate standard error of the mean. ** *p* < 0.01 and *** *p* < 0.001.

## Data Availability

The data presented in this study are available on request from the corresponding author.

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
