# Peer review of "The Benefit of Bimodal Training in Voice Learning"

_brainsci, 2023, doi:10.3390/brainsci13091260_

Round 1

Reviewer 1 Report

The authors initially trained four groups of participants to learn the voices of ten talkers presented alone, or together with their own articulating or static face, or isolated articulating mouth. Participants were then tested on recognizing the voices on their own regardless of their training modality. It was found that the voices learned with articulating or static faces were recognized better compared to voices learned alone. However, isolated mouth movements did not provide the same advantage in voice learning as articulating whole face; a finding which is in contrast with the expectation that the former condition would be as effective as the latter one. The study is interesting. Details related to the procedures and methods employed are neatly explained. The results are clearly presented and adequately discussed in the light of relevant literature. However, the following issue might be considered by the authors:

The authors mention in lines 379 and 522 that they performed some post-hoc slope analyses to compare the learning rates (how quickly participants learned the voices) in different conditions. However, it is not clear how these analyses were performed, what the dependent variable was, and where (in which figure) their results are documented (illustrated)?

 A typo:

 Line 322: “analysis if variance” should read “analysis of variance”.

Author Response

We would like to thank Reviewer 1 for providing valuable feedback. We appreciate their thoughtful comments, and we have addressed each of their concerns (including the typographical error) in the revised version of the paper.

In the revised version we have added the following statement about our post-hoc analysis. “To examine the learning rates, we ran another set of regression models in which condition, probe trials, and the interaction between condition and probe trials were treated as fixed effects and subjects and probe talkers were included as random effects. Accuracy on the probe trials were included as the dependent variable (Lines 404-407).” We hope this clears up any confusion.

Reviewer 2 Report

Zadoorian and Rosenblum explored whether the learning of voices (when tested on auditory only speech) was improved when previously presenting/training the voices with the auditory information alone, a static face, a face with congruent articulatory movement, or only the articulating mouth. The results seemed to imply that voice learning was best in for articulating or static faces, and that the articulating face did not provide any advantage. 

Overall I found the experimentation and analysis to be relatively strong, but did have several concerns. 

First and foremost, while the presentation of marginally significant findings was at some point acceptable, in our current era of open science this is generally frowned upon. Personally (perhaps because I'm older), I don't mind some degree of supposition in the discussion for these types of findings, but much of the interpretation of the results and the supporting discussion rely on these marginal findings. I think the paper would be stronger and more scientifically sound if they were treated as insignificant and the discussion modified accordingly. 

Other concerns:

- please put citations in parenthesis in alphabetical order (throughout manuscript)

- Consider writing more of the paper in past tense (e.g., the goal was rather than is)

- I'm sure the editors will help with this, but the paragraph indentations and formatting were off in the version of the manuscript that I received

- Consider including Sumby and Pollack (1954) and more recent research showing how visual information can improve the intelligibility of speech. There are many findings in the cross modal literature supporting this. 

- Lines 179-184 discuss research showing how direct gaze impairs performance. The present findings potentially show the opposite. How do the authors account for these differences?

- Why would online recruitment limit the number of participants? If anything, I expected the opposite? 

- Was a power analysis done to determine the number of subjects? Should one be done now just to show that the number of subjects was appropriate? 

- Consider a follow up analysis that does not include the non-native speakers. 

- Line 258 - "See Figure X". This figure is missing. Include figure and number as needed. 

- Perhaps change phrasing such as, "...there was not a a significant difference..." to something like, a significant difference was not observed....

- The discussion discusses the articulating face condition as moving faces. It might be clearer to use the terminology used in the intro rather than "moving". Moving seems to imply other facial movements rather than the articulation. 

- Could the increased amount of information in static/articulating faces conditions compared to just the mouth be the reason for the improved voice recognition? It's easy to bind? Or simply the 'weirdness' of only the mouth being presented? Could that harm identification? 

- line 322 - analysis of (not if)

- Hanging indents are missing in some of the citations. 

English is fine. Just needs some light editing throughout. 

Author Response

We would like to thank this reviewer for providing many important comments and suggestions for our paper. We have incorporated changes to address their points, including grammatical as well as typographical errors, and believe the paper has benefitted greatly from their input.

In the revised version (Lines 608-623) we changed the interpretation of the static face condition result from ‘marginally significant’, to ‘non-significant’—along with our interpretation— as per the reviewer’s request. However, we do still acknowledge that no statistically significant difference in performance was observed between the articulating face and static conditions, and that the static face condition did show a trend toward facilitating performance more than and audio-alone training conditions. Thus, while we soften our conclusions regarding the static face condition we do still acknowledge that static faces may have the potential to facilitate some voice learning. We also state that this will be addressed in follow up studies.

Next, the reviewer has commented on our number of subjects and whether it is sufficient for our study. We have now added a comment about this concern (Lines 222-228) and acknowledge that because the study was conducted during COVID quarantine, we were limited in the number of subjects we could run. As mentioned in the manuscript, while all of our subjects were run remotely, we did not feel comfortable letting our subjects ‘run themselves’ through online recruitment. We therefor chose to have lab researchers guide and monitor each of our 96 subjects through the experiment (via Zoom) to ensure that subjects understood the task and were paying attention appropriately. Thus, all subjects had to be scheduled based on finding a mutual 90 minute period that would work for each subject and experimenter. This severely constrained the number of subjects that could be run. Still, we were able to run 24 subjects per condition, which is comparable, or greater, than other related studies in the literature (e.g., Sheffert & Olson, 2004; von Kriegstein, et al. 2008).

Next, the reviewer recommended that we examine how native-language background played a role in our results (83 of 96 subjects were native English speakers).  Accordingly, a post-hoc analysis was conducted in which the main comparisons were analyzed after excluding the non-native speakers. In this regression analysis, accuracy on the test trials was included as the dependent variable, condition was treated as fixed effect and subject and talkers were included as random effects. Results of this analysis revealed a significant difference between the articulating face and audio-alone conditions (p = 0.0132). However, a significant difference was not observed for the static face (p = 0.1312) and isolated mouth only (p= 0.9365) conditions compared to the audio-alone condition. After removing the non-native speakers from the analysis, the mean performance during the audio-alone test trials were as follows: 67% for the articulating face (compared to 65% when the non-natives were included), 61% for the static face (compared to 63% when the non-native were included), 55% for the isolated mouth (compared to 53% when non-natives were included), and 55% for the audio-alone conditions. Mean values for the audio-alone condition did not change. Because excluding the non-native English speakers did not change the pattern of results, we have decided not to report this analysis in the paper.

In the revised version of the manuscript, we have added citations (including the Sumby & Pollack, 1954) stating how visual information can improve the intelligibility of speech (“It is well known that adding visual information facilitates intelligibility of speech [11, 38, 40].”) Additionally, the following footnote was added in the manuscript (page 3) addressing the differences between the current study and Lavan and colleagues (2023) showing how direct gaze impairs performance.

“1. Although it is true that in the present study, both the articulating and static face training conditions included a direct gaze of the speakers, it is important to note that the training included other characteristics that may have outweighed the effect. For example, in the present study, participants were presented with audio-alone probe trials during training to ensure they were learning the voices. Additionally, during the test phase, participants were asked to identify the talkers (compared to categorizing voices as old/new).”

Per the reviewer’s suggestion, we have changed the name of the moving face condition to ‘articulating face’ throughout the paper. We have also added Figure 1 illustrating the isolated mouth-only and static face stimuli.

Lastly, our results revealed a significant difference between the articulating face and isolated mouth conditions during the test phase, favoring the articulating face condition. Similarly, a marginally significant difference was observed in the static face and isolated mouth condition favoring the static face condition. These results could suggest that the increased amount of both articulatory and non-articulatory information improved voice recognition. As mentioned in the manuscript (Lines 572-579) the mouth only stimuli might have failed to capture enough important articulatory information available outside the mouth area (“The mouth-only stimuli only displayed a very small oval around the mouth-area (see Fig. 1). It is possible that these stimuli did not capture enough important articulatory information present outside the mouth area. It is known that important visible articulatory information is present throughout the face [e.g., 15, 44]. To test the possibility that the mouth-only condition provided too little articulatory information, follow up studies can be conducted in which additional articulatory information is visible including movements of the jaw and cheeks. ”). Additionally, another possible reason could have been the ‘weirdness’ of the isolated mouth stimuli (“Another possibility is that the visibly unusual nature of the mouth-only stimuli may have been distracting to the degree that it prevented sufficient attentional resources from being used for good voice encoding.” Lines 580-582).

Reviewer 3 Report

The findings will help researchers design different tool systems to assist those with hearing and language impairments. This study is valuable in terms of the results of the study. This study could have been more valuable if individuals with hearing impairment had been evaluated.

ıt is enough for Quality of English Language.

Author Response

We are grateful to this reviewer for the valuable feedback they have offered. We agree that a study involving a hearing-impaired population is a very good idea, and something we will certainly be considering in the near future. We now acknowledge this point and that we plan to address it in future research (Lines 632-633).

Reviewer 4 Report

The text of the paper is not formatted according to the standards of a scientific publication. The introduction is missing while it's been replaced with a section "The Benefit of Bimodal Training in Voice Learning" which is too long and not focused on the aim of the paper. The paper is too long and not focused. The introduction should be included with very short background and review of previous work on the topic. Aims and novelties of the study should be clearly described as well as potential relations with previous studies.

Method section contains subsections introduced in an inadequate manner. 

Overall, this paper in the present form is unfit both for serious review or potential publication. The authors should write adequate introducton and shorten the text significantly and format the paper according the the journal requirments. 

Author Response

We would like to thank this reviewer for taking the time to review our manuscript. We have made every effort to thoroughly explore and incorporate studies that directly relate to our research objectives.

We believe we have addressed the aims of the study under the heading titled “Purpose of the Current Study.” (Lines 181-220). In response to their feedback regarding the method section, we have taken great care to address every relevant detail pertaining to the stimuli used in our study, as well as detailed description of the study procedure. We have attempted to provide a clear description to enable fellow researchers to replicate our experiments with accuracy.

While the reviewer commented on the paper not being appropriate for the journal, we had asked the editor of the special issue whether our paper was appropriate before submitting, and were told that it was. We will therefore leave this decision to the editors.

Round 2

Reviewer 2 Report

This is the second time I have reviewed this paper and I found it to be much improved. I appreciate the effort and diligence that the authors demonstrated in their revision. 

I believe that the paper is suitable for publication, but do have two final comments. 

* I appreciate the significant effort and time it took the experimenter to meet with participants over zoom. We did this exact same thing in my lab and indeed, it was taxing to say the least. Nonetheless, I would suggest the inclusion of a power analysis just to demonstrate that the number of subjects is adequate for this study. While it does align with previous research and such an analysis is after the fact, best practices would suggest that a power analysis is needed.

* Thank you for conducting the analysis excluding non-native speakers. The authors might want to include a footnote or comment simply stating that this analysis was conducted and that the results were the same. I think that would make the paper stronger and some readers might appreciate that. 

Author Response

We are grateful that this reviewer provided a second round of valuable feedback. Two footnotes have been added to page 5 addressing: a) a report of the post-hoc power analysis and; b) findings of the critical analysis excluding the non-native speakers.  

  • “The number of subjects tested was based on a) a similar number used in related experiments (e.g., Sheffert & Olson, 2004); and b) the limitations of running monitored subjects online during COVID quarantine. As per a reviewer’s suggestion, a post hoc power analysis was conducted using the R package “simr” [13]. The effect size was calculated using the doTest function from the “lmertest” package [17]. The effect size calculation was based on 50 simulations using the z-test and the alpha level was set to 0.05. The effect size for the articulating face condition was equal to 0.61 with an observed power of 70% (CI: 55.39, 82.14). The effect size for the static face condition was equal to 0.43 with an observed power of 38% (CI: 24.65, 52.83). Lastly, the effect size for the isolated articulating mouth condition was equal to -0.019 with an observed power of 8% (CI: 2.22, 19.23).”
  • “The critical analysis (comparing the test means between the four conditions) was repeated after excluding the non-native speakers, and the analysis yielded the same results.”

All changes are highlighted in the newly submitted version.

Reviewer 4 Report

Based on the fact that the article is too lengthy and the formating is confusing. I thought it is the authors' duty to follow the formating guidelines before submission. In this manner valuable reviewers' time is not wasted on reading the material which is not suitable for review or publishing. Currently I don't have other concerns.

Author Response

We would like to thank this reviewer for reading the manuscript for the second time. Per the editor’s suggestion, we have changed the format of our manuscript.

All changes are highlighted in the newly submitted version.

Round 3

Reviewer 4 Report

Authors have improved the manuscript based on my suggestions

Author Response

All editor comments have been incorporated. Thanks!